# Static and Dynamic Properties Study on Interface between New Polymer Materials and Silty Clay Based on Ring Shear Tests

**DOI:** 10.3390/polym15030634

**Published:** 2023-01-26

**Authors:** Jia Li, Jie Li, Jingwei Zhang, Guangzong Liu

**Affiliations:** School of Water Conservancy and Civil Engineering, Zhengzhou University, Zhengzhou 450001, China

**Keywords:** polymer anti-seepage wall, polymer materials, original designed test mould, ring shear tests, static and dynamic properties of the interface, the hyperbolic constitutive model

## Abstract

The polymer anti-seepage wall composed of polymer materials is a new technology for impermeable reinforcement in dykes and dams. Compared with traditional grouting materials, polymer grouting materials have the advantages of early strength, convenience, good anti-seepage performance, safety and durability. Because of the particularity of polymer materials, they form a “root-like” cementing status with dam soils after grouting. This complex interface affects the interaction between the wall and the dam, which subsequently influences the whole structure’s properties under loads. In this paper, based on an original designed test mould, an SRS-150 dynamic ring shear instrument was used to conduct static and dynamic ring shear tests to explore the static and dynamic properties of the polymer–silty clay interface. Moreover, influence laws and the related mechanism of different factors on the polymer–silty clay interface were studied in this paper. At the same time, the hyperbolic constitutive model of the polymer–silty clay interface was established, and the validity of the model was verified by comparing the numerical simulation with the relevant experimental results. The achievements of this paper are helpful as they provide a scientific basis for the structure’s mechanical analysis and lay the foundation for the promotion and application of the new anti-seepage technology.

## 1. Introduction

The polymer anti-seepage wall is a new technology that has developed in recent years for the anti-seepage reinforcement of dams [1]. It has the advantages of a fast speed, being lightweight, high toughness, economy, durability and environmental protection, etc., and meets the urgent need for dam reinforcement. It has gradually become the main measure for the anti-seepage reinforcement of medium and small reservoirs and earth–rock dams [2,3,4,5,6,7,8,9]. To date, some scholars have studied the degradation of polymer grouting materials. Shi [10] studied the corrosion resistance and aging resistance of polymer grouting materials. When it is in the underground environment for a long time, the polymer grouting material does not easily degrade or lose its stability due to the chemical corrosion in the normal foundation environment. Hanover University [11] used differential thermal analyses to quantify the thermal stability of the polymer materials as a loss of quality, and conducted a long-term continuous test on polymer grouting materials. The results show that the life of the material can reach 33 years, and more than 100 years when used underground. The above experiments are sufficient to prove that polymer grouting materials have long-term stability and do not cause degradation problems that adversely affect the surrounding environment. The polymer grouting technology is used to grout non-water reactive polymer materials into the place reserved inside the dam body at a high pressure through professional grouting equipment, and then the polymer materials undergo a chemical reaction and expand rapidly in volume filling the cavity and solidifying to form a polymer anti-seepage wall [12] (Figure 1).

The non-water reactive polymer grouting materials are composed of raw materials such as polyisocyanates and polyester polyols or polyether polyols, which have advantages of good anti-seepage ability, good ductility, good tensile and compressive capabilities, good expansion force and excellent durability [13]. At the same time, the solidified body of the polymer grouting material after the reaction is not degraded, and is pollution-free and has good stability. Therefore, polyurethane polymer grouting materials as ideal chemical grouting materials have been widely used in the field of civil engineering to repair or reinforce engineering diseases, including for highways, bridges, tunnels, reservoirs, dams, coal mines [14], geological exploration and other projects in waterproof plugging, building foundation reinforcement, complex underlying stability, road pavements and airport pavement maintenance [12,15,16]. Among them, the cumulative application mileage of polymer grouting technology in expressway rapid repair has reached 3 million km, and the cumulative repair disease area has exceeded 1 million m^2^. The complete set of polymer grouting technology for dam anti-seepage reinforcement has been applied in more than 200 dangerous dam reinforcement projects and more than 20 small and medium-sized river management projects [1]. As a new anti-seepage wall reinforcement technology, when the polymer grouting material is grouted into the soil, it forms a “root-like” structure that infiltrates into the soil, as shown in Figure 1a, which complicates the wall–dam interface and makes it differs from the static and dynamic properties of the interface between the traditional concrete anti-seepage wall and dam. However, the interface has a great influence on the load transmission and deformation coordination between the soil and the structure, and is also a weak point in the soil–structure interaction system, which affects the structural stability of the anti-seepage wall [17,18]. In view of this, it is essential to conduct research on the static and dynamic properties of the interface between the polymer anti-seepage wall and the earth–rock dam.

The shear properties of the soil–structure interface form the basic theoretical bases for studying the interaction between soil and structures [19]. Many domestic and foreign scholars have performed a series of work on the experimental study of the shear properties of soil–structure interfaces. Aimed at the influencing factors of the shear properties of the interface, Feng et al. [20] conducted an interface cyclic shear test by using an 80 t large three-dimensional interface cyclic single shear tester to explore the impact of vertical stress on the properties of the interface, and the results showed that the vertical stress had a significant effect on the tangential stress and tangential stiffness of the interface; Liu et al. [21] conducted a shear test on the interface between the wall and dam by considering the effects of different levels of interface roughness, different vertical stresses and different soil sample thicknesses on the interface properties in the test, and simultaneously defined the concept of “nominal interface”; Liang et al. [22] investigated the factors influencing the interface surface properties of the silty clay structure by using the ring shear instrument and proved that the interface properties developed from strain softening to strain hardening when the vertical stress was greater than 100 kPa. Meanwhile, they obtained parameters such as adhesive force and internal friction angle of the interface. For the study of interface shear deformation, Kishida et al. [23] first used a stacked ring single shear instrument to conduct shear tests on the interface of the soil–structure and found that large shear deformation occurs at the shear zone of the interface; Bishop et al. [24] found that the rupture surface occurred at the junction of the upper and lower shear boxes by using a new ring shearing instrument to perform ring shear tests on different clay soils. Aiming at the development law of the shear properties of the interface, Wu et al. [25] studied the dynamic properties of the interface such as shear stiffness and damping ratio by using a single shear instrument, and the test results showed that the dynamic shear stress and relative displacement of the interface of the soil and concrete exhibited a hyperbolic relationship; Yang et al. [26] used the HJ-1 ring shearing instrument to conduct ring shear tests on the interface between clay and clay core dams, and found that the mechanical properties of different types of interfaces differed greatly. The interface of the clay core dam exhibits properties of strain hardening, and the stress–strain trend conforms to the hyperbolic relationship.

The shear stress–shear displacement curves between the soil and the structure can reflect the deformation law of the interface and the reliability of engineering design is closely related to the selected constitutive model [27]. The constitutive model of the interface between the soil and the structure has been studied by many scholars. At present, the nonlinear elastic model based on the hyperbolic model proposed by Clough et al. [28] is widely used [29]. In addition, Boulon et al. [30] proposed an elastic–plastic model of interface deformation based on the relationship between shear stress and relative displacement in the direct shear test, and its tangential constitutive relation is similar to the hyperbolic model. Esterhuizen et al. [31] used the shear test of the clay–geotechnical material interface to obtain that the shear stress of the reinforced soil interface decreases significantly after reaching the peak value, showing the phenomenon of shear softening, and proposed a hyperbolic model to fit the nonlinear properties after the peak value. Liu et al. [21] mainly investigated the influence of roughness, normal stress and soil sample thickness on the shear properties of the interface by using the improved cyclic simple shear testing system, and proposed that obtaining the shear strength properties and parameters of the interface is a key issue in the study of the constitutive relationship of the interface.

However, most of the above studies were aimed at the interface mechanical properties of soil–traditional geomaterials, and the shear properties of the interface between polymer grouting materials and the earth–rock dam have not yet been thoroughly and systematically researched. Therefore, according to the complex stress state of the interface between a polymer anti-seepage wall and a dam, the static shear test and cyclic shear tests of the polymer–soil interface were carried out to study the influence laws and related mechanism of different factors on the static and dynamic properties of the polymer–soil interface. At the same time, the hyperbolic constitutive model of the polymer–soil interface was established and verified by comparing the numerical simulation with the relevant experimental results. The research results are expected to provide scientific basis and theoretical guidance for the further promotion and application of the new anti-seepage technology.

## 2. Ring Shear Test Research on Polymer–Soil Interface Properties

### 2.1. Test Equipment

The polymer–soil interface ring shear test adopts the SRS-150 dynamic ring shear instrument of GCTS Company in the United States, and the main parameters of the SRS-150 dynamic ring shear instrument test device are shown in Table 1. The SRS-150 dynamic ring shear device consists of three parts: the main ring shear module, the PCP-15U pressure plate and the DA/PC system, as shown in Figure 2.

### 2.2. Mould Design

Based on the “root-like” contact state of the polymer–soil interface and the liquidity and expansion of the polymer materials, a special mould is designed to fabricate the polymer–soil sample, as shown in Figure 3. The mould consists of a fixed structure and a mould-forming structure. The fixed structure includes a bottom plate, an upper cover, a screw, a pressure relief cover plate and an injection cover plate, while the mould-forming structure includes a core body and a ring. The bottom plate and the upper cover, the pressure relief cover plate and the injection pressure plate are all connected by bolts, respectively, and the ring is placed inside the core body. In order to avoid the damage caused by the grouting pressure to the soil, a new type of grouting duct (Figure 4) needs to be added, which is designed as follows: the upper part of the grouting duct is connected to the polymer grouting port, and the lower part is embedded in the ring body. The polymer slurry flows into the grouting duct from the grouting port and is sprayed on the wall of the circle at an angle of 20°, which dissipates the grouting pressure and effectively avoids the damage to the soil sample caused by the polymer slurry under high pressure.

The mould assembly steps are as follows: Firstly, place the soil sample in the core body. Secondly, assemble and fasten the fixed structure and the mould-forming structure. Then grout the polymer slurry from the grouting port into the grouting pipe through the pressure equipment. When the polymer slurry comes into contact with the soil sample, the reaction of coagulation and solidification takes place. Finally, the grouting ends when the polymer slurry overflows from the slurry outlet.

### 2.3. Preparation of Samples

In the test, the two-component isocyanate and polyol materials are preheated using polymer integrated grouting apparatus, and the polymer samples are prepared by the constant pressure grouting device in a certain proportion. At the same time, the density of the specimens can be controlled by injecting different masses of polymers into a fixed volume of the mould. The non-water reactive polymer grouting materials used in the test were provided by Zhengzhou Anyuan Engineering Technology Co., Ltd., Zhengzhou, China, and the technical indicators are shown in Table 2. Additionally, the test chose silty clay as the dam material, and the parameters of its main physical and mechanical properties are shown in Table 3. The specific preparation process of the polymer–soil interface sample (Figure 5) is as follows: ① filling the soil sample in the core body; ② daubing lubricating fluid to the ring; ③ fastening the connection between the core and the ring; and ④ grouting in the mould with grouting equipment. After grouting, the samples were demoulded after standing for 1.5 h at normal atmospheric temperature, and all the prepared samples met the requirements of the polymer–soil interface ring shear test. The sample after grouting is shown in Figure 6.

### 2.4. Experimental Methodology

The program of this test is divided into two parts: the static shear test program and the cyclic shear test program. The static shear test program mainly takes into account some influence factors such as the bonding state, different polymer densities, vertical stresses, different dam materials and different anti-seepage bodies, etc., as shown in Table 4, while the influence factors of the cyclic shear test include different polymer densities, different shear stress magnitudes and different vertical stresses as shown in Table 5.

## 3. Analysis of Static Test Results of Polymer–Soil Interface

### 3.1. Influence of Different Bonding States

When we directly grout the polymer on the silty clay, the polymer has not been consolidated at this time, showing a certain flow diffusivity, with strong adhesion and bonding ability. At this time, the polymer fills in the gap between the adjacent soil particles, and the adjacent soil particles are adhered to form the bonding state of the interface. When waiting until the polymer solidification is formed and then combined on the soil, it is in a non-bonded state.

In the case of a vertical stress of 100 kPa and polymer density ρ = 0.188 g/cm^3^, the shear stress–shear displacement relationship curves of polymer–silty clay interfaces with different bonding states are plotted in Figure 7, and the shear strength indexes of the interfaces with different bonding states are presented in Table 6.

From Figure 7 and Table 6, it can be seen that with the increase in shear displacement, the shear stress of the polymer–soil interface in the bonded condition continues to increase, while the shear stress of the interface in the non-bonded condition reaches a peak at 10 mm and then decreases slightly before reaching a steady state, which shows the characteristic of shear softening. At the same time, the peak shear stress of the interface in the bonded condition is increased by 3.4 kPa compared with that in the non-bonded condition; the internal friction angle and the adhesive force of the polymer–silty clay (bonded) interface are increased by 0.42° and 4.2 kPa, respectively, compared with those of the polymer–silty clay (unbonded) interface.

The reasons for the above phenomenon can be summarized as follows: the bonding function causes the polymer–silty clay (bonded) interface’s internal friction angle and adhesive force to increase compared with those of the polymer–silty clay (non-bonded) interface, which enhances the interface’s peak shear stress in the bonded condition correspondingly. In addition, when the shear test is performed, the adhesive force of the polymer–silty clay (bonding) interface increases constantly due to the permeation and bonding of the polymer grouting material, which enhances the degree of hardening of the stress–strain curve. However, the polymer–silty clay (non-bonded) interface lacks adhesive force, and the shear stress–displacement curve produces a slight strain softening phenomenon after reaching the shear stress peak. Therefore, in order to make the calculation results more realistic, the bonding function of the polymer–silty clay interface should be considered in the analysis of the interaction between the wall and the dam in the earth–rock dam project.

### 3.2. Influence of Polymer Density

Under the condition of 100 kPa vertical stress, the shear stress–shear displacement relationship curves of the polymer–silty clay interface under different densities are plotted in Figure 8, and the shear strength indexes of interfaces with different polymer densities are shown in Table 7.

As can be seen from Figure 8 and Table 7, with the polymer density increasing from 0.188 g/cm^3^ to 0.280 g/cm^3^, the peak shear stress at the polymer–silty clay interface increases continuously, and the adhesive force and internal friction angle of the polymer–silty clay interface also increase successively. At the same time, the shear stress under different polymer density conditions all increase with the increase in shear displacement.

Analysing the reasons for the above phenomena, according to the research results of Zheng et al. [32], it is known that the bond strength of polymer materials is positively correlated, and the increase in bond strength makes the bonding function between the interfaces more significant, which improves the shear resistance of the interface. At the same time, the increase in polymer density improves the expansion anchoring force of the polymer materials, which enhances the shear strength of the polymer materials. In view of this, as the density of the polymer increases, its effect on the mechanical properties of the interface becomes more and more obvious, so the choice of polymer density has a significant effect on the grouting effect in the actual grouting process. Actually, it does not mean that the mechanical properties of the interface can be improved simply by increasing the polymer slurry density. The optimal polymer slurry density should be considered by taking into account the actual soil loading circumstances, anticipated improvement effect, economy and other considerations, comprehensively.

### 3.3. Influence of Vertical Stress

The shear stress–shear displacement relationship curves of the polymer–silty clay interface under different vertical stresses are presented in Figure 9.

As can be seen from Figure 9, in the case where there is the same density of polymer, the slope of the initial section of the curve increases with the increase in vertical stress, which means that the initial shear modulus of the polymer–silty clay interface is positively correlated with the vertical stress. Meanwhile, when the vertical stress increased sequentially from 100 kPa to 400 kPa, the peak shear stress increased from 60 kPa to 208.1 kPa, showing shear hardening properties.

It is known from the test that with the increase in vertical stress, the compactness of the soil near the interface of polymer–silty clay also increases, and the interaction between the polymer and the soil becomes closer, which improve the adhesive force of the interface. At the same time, when the vertical stress increases, the frictional resistance between the soil particles also increase accordingly, which drives more soil particles to flip, roll and rearrange and increases the peak shear stress of the interface. At this point, a larger load needs to be applied to make the interface if shear damage is to occur, and the interface strain hardening tendency becomes more pronounced.

### 3.4. Influence of Dam Construction Materials

The test results of the interface shear stress–shear displacement relationship curves between different dam-building materials and polymer are shown in Figure 10. The shear strength indexes of the interfaces of different dam-building materials are shown in Table 8.

From Figure 10 and Table 8, it is clear that the peak shear stress of the polymer–silty clay interface is 4.9 kPa higher than that of the polymer–silty clay interface under the same polymer density, which means the polymer–silty clay interface has better shear properties than the polymer–silty clay interface. In addition, it can be seen that the adhesive force and internal friction angle of the polymer–silty clay interface are increased by 1.3 kPa and 4.1°, respectively, compared with those of the polymer–silty clay interface.

As for the causes of the above phenomenon, the following analysis can be made: under the same test conditions such as the water content of soil samples, the adhesive force and internal friction angle of silty clay are larger than those of silty soil, which means the shear strength of the silty clay is better than that of the silty soil. Meanwhile, the content of clay particles in silty clay is also higher than that in silt. The presence of clay particles enhances the adsorption of the interface and the nearby soil, which improves the peak shear stress of the interface. Therefore, the shear resistance of the polymer–silty clay interface is better than that of the polymer–silt interface. In the actual earth–rock dam project, the dam construction materials containing clay particles can be considered to fill the dam body so as to obtain better structural shear performance.

### 3.5. Influence of Different Anti-Seepage Bodies

The test results of the shear stress–shear displacement relationship curves of the interfaces between different anti-seepage bodies and soils are shown in Figure 11. The shear strength indexes of the interfaces of different anti-seepage bodies are shown in Table 9.

From Figure 11 and Table 9, it can be seen that under the same vertical stress, the polymer–silty clay interface shows a trend of strain hardening, and the shear stress increases continuously with the shear displacement. However, the concrete–silty clay interface shows a tendency of strain softening, and the shear stress reaches its peak and then begins to decrease and finally tends to a stable residual strength. Moreover, the adhesive force of the polymer–silty clay interface is higher than that of the concrete–silty clay interface, and the difference is 3.55 kPa.

Analysing the essence of the phenomenon, it can be shown that when the adhesive and consolidation effects occur between the polymer and the soil, the polymer is filled in the gap between adjacent soil particles after grouting to form a “root-like” structure, which adheres to the adjacent soil particles, improves the compactness of the soil and achieves the effect of solidifying the soil, thereby enhancing the shear resistance of the interface. At the same time, the shear deformation of the sample is mainly caused by the fragmentation and tumbling arrangement of the soil particles near the interface. When the morphology of soil particles is disrupted, more angularities are created on the surface, so it can bite better with the polymer (Figure 12), which makes the shear strength of the interface increase in the process of shear deformation and show strain hardening properties. Additionally, under the vertical stress of 100 kPa, the shear strength of the concrete–silty clay interface is smaller than that of the silty clay–silty clay, so the shear deformation of the sample is caused by the misalignment slip of the concrete and soil interfaces. When the misalignment slip occurs, the bonding state of the polymer–soil interface is destroyed, which makes the shear strength of the interface decrease during the shear deformation of the sample and show strain softening properties in the shear process. Therefore, the bonding effect of non-water reactive polymer grouting material–soil is better than that of conventional concrete material–soil.

### 3.6. Influence of Additional Polymer Anti-Seepage Wall in the DAM Body

The polymer anti-seepage wall is a subsidiary structure formed by grouting in the dam body after earth–rock dam construction. Comparing the shear test results of polymer–silty clay and plain soil is of great significance to analyse the influence of the polymer anti-seepage wall on the structural stability of the dam body.

The relationships of the shear stress–shear displacement curves of plain soil and polymer–silty clay are shown in Figure 13, and the shear strength indexes of their interfaces are presented in Table 10. From Figure 13 and Table 10, it can be seen that the shear stress–shear displacement curves of both polymer–silty clay and plain clay exhibit strain hardening under the same vertical stress, while the shear stress at the interface of polymer–silty clay is increased by 11.2 kPa compared with that of plain soil. At the same time, it is clear that the shear strength indexes at the interface of polymer–silty clay are higher than those of plain soil, and the adhesive force and internal friction angle at the interface of polymer–silty clay are increased by 0.42 kPa and 0.71°, respectively, compared to those of plain soil.

The reasons for the above phenomenon can be summarized as follows: due to the similar elastic modulus of polymer materials and silty clay, polymer materials and silty clay exhibit good deformation coordination ability, which makes the adhesive force develop steadily under the vertical stress. At the same time, the plain soil changes from loose to dense when subjected to the vertical stress, and the adhesive force of the interface also maintains a stable development, so the strain hardening properties of polymer–silty clay and plain soil are similar. However, compared with silty clay, the expansion extrusion and the permeation bonding of the polymer increase the shear strength of the polymer–silty clay interface, which improve the shear properties of the contacting silty clay and increase the anchoring force of the soil near the interface. Therefore, the construction of polymer anti-seepage walls in the dam body can not only enhance the anti-seepage performance of earth–rock dams but also improve the overall stability and shear resistance of the earth–rock dams.

## 4. Analysis of Dynamic Test Results of Polymer–Soil Interface

The hysteresis curve is formed by the round-trip development of shear stress and shear displacement at the interface of polymer–silty clay under cyclic loading. The hysteresis curve represents the energy absorption and dissipation process of the polymer–silty clay interface, and the area of the hysteresis curve represents the magnitude of energy absorbed by the polymer–silty clay interface. Through the cyclic shear test on the interface of polymer–silty clay, the effects of polymer density, shear displacement amplitude and vertical stress on the dynamic shear stress–shear displacement of the interface can be obtained, and on this basis, the development and change law of the interface shear stiffness and damping ratio can be summarized. The shear stiffness is one of the parameters reflecting the resistance of the interface to shear deformation damage, and the damping ratio reflects the change regularities about the energy dissipation of the interface under the dynamic action [33,34,35].

### 4.1. Influence of Polymer density

The dynamic shear stress–shear displacement relationship curves for the 10th cycle shear of the interface for three conditions with polymer densities of 0.188 g/cm^3^, 0.210 g/cm^3^ and 0.280 g/cm^3^, respectively, are shown in Figure 14. The test results of the shear stiffness development trends of the interface under different densities are shown in Figure 15, and the damping ratio development trends are shown in Figure 16.

As can be seen from Figure 14, with the increase in polymer density, the hysteresis loop image of the polymer–silty clay interface begins to expand. As the hysteresis loop area increases, the peak shear stress of the interface also increases. At this time, the peak shear stresses corresponding to the polymer densities of 0.188 g/cm^3^, 0.210 g/cm^3^ and 0.280 g/cm^3^ are 28.9 kPa, 35.4 kPa and 56.8 kPa, respectively. The effect of polymer density on the interfacial shear strength is largely through increasing the shear modulus and shear strength of the polymeric material to increase the peak shear stress strength at the polymer–clay contact interface.

As can be seen from Figure 15, when the number of cyclic shearings increases, the shear stiffness of the polymer–silty clay interface increases with the increase in the number of cycles, and decreases slightly after reaching the peak value then tends to develop steadily. The reason for this phenomenon may be related to the breakage of soil particles on the interface after cyclic shearing and the change in the interfacial shear strength caused by the rearrangement of the particles. In addition, as the polymer density increases, the shear stiffness of the interface also increases. With the increase in the density of the polymer materials, the shear modulus and shear strength of the polymer increase, and the permeation and bonding function of the interface is also improved accordingly, which improves the shear stiffness of the interface.

From Figure 16, it can be seen that the damping ratio of the interface decreases with the increase in the number of cycles and the density of the polymer. The reason for this phenomenon, according to the group’s previous study [36], is that the increase in polymer density improves the storage modulus of the material, reduces the loss factor, slows down the energy dissipation rate of the polymer–silty clay interface, and intensifies the rearrangement of particles at the interface in the cyclic shear test. Therefore, the damping ratio of the polymer–silty clay interface decreases with the increase in the number of shear cycles.

### 4.2. Influence of Shear Displacement Amplitude

When the shear amplitude γ is set to 0.5 mm, 1 mm and 1.5 mm, the dynamic shear stress–shear displacement relationship curves of the polymer–silty clay interface for the 10th cycle are shown in Figure 17, the shear stiffness development trends of the interface are plotted in Figure 18 and the damping ratio development trends are presented in Figure 19.

As can be seen from Figure 17, the hysteresis loop curve gradually develops from a regular ellipse to an S-shape when the shear displacement amplitude increases from 0.5 mm to 1.5 mm, and the areas of the hysteresis loop and the peak shear stress increase accordingly. The reason for this phenomenon is that as the shear displacement amplitude increases, the soil particles near the polymer–silty clay interface become more orderly and the energy absorbed and dissipated increases, which further increases the shear stress and improves the shear strength of the interface.

As shown in Figure 18, the shear stiffness of the polymer–silty clay interface increases with the increase in shear displacement amplitude. However, at lower shear displacement amplitudes, the interfacial shear stiffness keeps increasing with the number of cycles, showing shear hardening properties, while at larger shear displacement amplitudes, the interfacial shear stiffness reaches a peak and then decreases, indicating that the shear hardening–softening alternation phenomenon exists at the interface during the cyclic shear. The reason for this phenomenon may be that when the shear displacement amplitude is small, the particles do not break significantly, and the dense effect of the cycle makes the interface contact more closely, which improves the shear stiffness continuously; meanwhile, when subjected to larger shear displacement amplitudes, with the increase in the number of cycles, a shear fragmentation zone appears at the interface and the particles at the interface undergo violent reorganisation, which causes the shear stiffness decreases.

According to Figure 19, the damping ratio of the polymer–silty clay interface is significantly sensitive to the shear displacement amplitude, and the damping ratio corresponding to the same number of cycles is positively correlated with the shear displacement amplitude, indicating that the increase in displacement amplitude leads to greater energy dissipation at the interface during cyclic shear. At the same time, it can also be seen that the interface damping ratio decreases and then increases with the number of cycles under the shear displacement amplitudes of 1.0 mm and 1.5 mm, while the damping ratio decreases with the number of cycles when the shear displacement amplitude is 0.5 mm.

### 4.3. Influence of Vertical Stress

Under the 100 kPa, 200 kPa and 300 kPa vertical stress conditions, the dynamic shear–shear displacement relationship curves of the polymer–silty clay interface for the 10th cycle are plotted in Figure 20, the shear stiffness development trends for vertical stresses are presented in Figure 21, and the damping ratio development trends are shown in Figure 22.

As shown in Figure 20, with the vertical stress increasing, the area of the hysteresis loop increases continuously, and the shear stress of the interface also increases. At this time, the shear stress increments are 3.1 kPa and 11.5 kPa, respectively. The reason for this phenomenon, on the one hand, is mainly due to the linear relationship between vertical stress and shear strength at the interface [32], the greater the vertical stress applied, the greater the frictional restraint between the soil particles near the interface will be, which contributes to improve the peak shear stress. On the other hand, as the vertical stress continues to increase, the pore structure of the soil changes and the soil near the interface is squeezed more compactly, while the soil particles are broken and the degree of shear hardening becomes more obvious, which leads to an increase in the increment of shear stress.

As can be seen from Figure 21, the shear stiffness of the interface develops with the increase in the number of cycles under the vertical stresses of 100 kPa, 200 kPa and 300 kPa, and the shear stiffness corresponding to the same number of cycles also becomes larger with the applied vertical stress increases. This is due to the increase in vertical stress that causes the stronger occlusal effect of the soil near the interface, which leads the soil to become more dense and improves the interface shear resistance, and shear stiffness is an important parameter characterizing the interface shear resistance to shear damage; thereby, the shear stiffness will also increase. It follows that an increase in vertical stress will lead to an increase in the shear stiffness of the interface within a certain range.

From Figure 22, it is clear that the damping ratio corresponding to the same number of cycles decreases with the increase in the vertical stress. At the same time, the damping ratios of the interfaces under the three vertical stresses is quite different in the initial several shear cycles. With the increase in the number of cycles, the development of the interface damping ratio under the three vertical stresses tends to be stable, indicating that the energy dissipation at the soil interface under cyclic shear tends to be stable with the increase in the number of cycles without considering the change in the physical state of the interface such as particle breakage, material deformation and wear under high vertical stress.

## 5. Nonlinear Constitutive Model of Polymer–Soil Interface

### 5.1. Static Interface Model

It can be seen from the test results that the stress–strain curve of the interface between the polymer anti-seepage wall and the earth–rock dam conforms to the hyperbolic relationship. The tangential and normal relationships of the nonlinear interface elements are expressed in matrix form as follows:(1)$$\left\{\begin{array}{c} \Delta \tau_{1}\\ \Delta \tau_{2} \end{array}\right\}=\left[\begin{array}{cc} k_{s1} & 0\\ 0 & k_{s2} \end{array}\right]$$

The nonlinear hyperbolic model of shear stress–shear displacement proposed by Clough and Duncan [28] is used to represent the stress–strain relationship of the nonlinear interface element, in which the asymptotic value of the hyperbolic is $(C-f\sigma_{y^{\prime }})/R$, and the tangential stiffness $k_{x^{\prime }}$ and normal stiffness $k_{y^{\prime }}$ are expressed in the form of quadratic parabola, as shown in Equations (2) and (3).
(2)$$k_{x^{\prime }}=k_{x^{\prime }}{}_{0}{\left[1-\frac{1}{\cos\alpha }\frac{R\tau_{x^{\prime }}}{\left(C-f\sigma_{y^{\prime }}\right)}\right]}^{2}$$
(3)$$k_{y^{\prime }}=k_{y^{\prime }}{}_{0}{\left[1-\frac{1}{\sin\alpha }\frac{R\tau_{y^{\prime }}}{\left(c-f\sigma_{y^{\prime }}\right)}\right]}^{2}$$

In the formula: $k_{x^{\prime }}{}_{0}$ and $k_{y^{\prime }}{}_{0}$ are the initial tangential stiffness and initial normal stiffness of the interface, respectively;

*C* is the adhesive force;

*ƒ* is the friction coefficient;

*α* is the friction angle of the shear surface.

Based on the nonlinear hyperbolic model proposed by Clough and Duncan [28], the shear deformation increment is replaced by the tangential and normal directions of the interface, which can be expressed as:(4)$$k_{sx}={\left(1-R_{f}\frac{\tau_{1}}{\sigma_{n}\tan\delta }\right)}^{2}K_{x}\gamma_{w}{\left(\frac{\sigma_{n}}{P_{a}}\right)}^{n}$$
(5)$$k_{sy}={\left(1-R_{f}\frac{\tau_{2}}{\sigma_{n}\tan\delta }\right)}^{2}K_{y}\gamma_{w}{\left(\frac{\sigma_{n}}{P_{a}}\right)}^{n}$$

In the formula: $K_{x}$ and $K_{y}$ are the tangential and normal stiffness coefficient, respectively;

Rf is the interface damage ratio;

δ is the friction angle of the interface;

$\gamma_{w}$ is water unit weight;

$P_{a}$ is atmospheric pressure.

Referring to the static shear test results of the polymer–silty clay interface, the tangential and normal stiffness coefficients are obtained according to the logarithmic slope of the stress–strain curve of the interface; the failure ratio $R_{f}$ is the ratio of the shear strength $\tau_{f}$ of the interface to the ultimate shear stress $\tau_{uk}$; and the friction angle of the interface δ is obtained by fitting the shear strength with the vertical stress curve. The tangential stiffness $k_{sx}$ and the normal stiffness $k_{sy}$ can be calculated by substituting the tangential stiffness coefficient $K_{x}$ and the normal stiffness coefficient $K_{y}$, the failure ratio $R_{f}$ and the friction angle δ of the polymer–silty clay interface into Equations (4) and (5). Then the shear stress of the interface will be updated automatically with the displacement variable iteration to obtain the stress–strain relationship of the interface. The interface model parameters are shown in Table 11.

### 5.2. Dynamic Interface Model

According to the results of the cyclic shear test, the shear stiffness and damping ratio are used to simulate the dynamic properties of the interface between the wall and the dam. The dynamic constitutive model of the interface is as follows:(6)$$K=\frac{K_{\max}}{1+\frac{K_{\max}R_{f}u}{\sigma_{n}\tan\delta }}$$
where $K_{\max}$ is the shear modulus;

$\sigma_{n}$ is vertical stress;

Rf is the interface damage ratio;

δ is the friction angle of the interface;

u is the shear displacement.

The damping ratio of the cyclic shear test results of the interface represents the energy dissipation of the interface, and its mathematical model [33] is as follows:(7)$$a=1/K_{1}\gamma_{w}{\left(\sigma_{n}/P_{a}\right)}^{n_{1}},b=R_{f}/\sigma_{n}\tan\delta$$
(8)$$\lambda =\frac{\lambda_{0}}{1+ku}+\left(\lambda_{ult}-\frac{\lambda_{0}}{1+ku}\right){\left(1-\frac{a}{a+bu}\right)}^{\alpha_{1}+\alpha_{2}\sigma_{n}}$$

In the formula: $\lambda_{0}$ is the initial damping ratio;

$\lambda_{ult}$ is the limit damping ratio;

$k,\alpha_{1}$ and $\alpha_{2}$ are the model parameters.

According to the cyclic shear test results of the interface, the dynamic parameters are obtained on the basis of the static interface model parameters, and the dynamic model parameters of the interface are shown in Table 12.

### 5.3. Implementation of Interface Unit

Due to the shear hardening properties of the interface between the polymer and silty clay, the nonlinear interface element can be used to simulate the shear hardening properties of the interface in the finite element analysis. The element is based on the starting point stiffness method and can simulate complex nonlinear contact problems through the FRIC subroutine interface provided by ABAQUS software [37,38].

The basic principle of the initial point stiffness method is to divide the shear stress into several incremental steps during the application of shear stress, and the shear stiffness coefficient is determined according to the initial shear stress of each shear stress increment step. On this basis, the FRIC subroutine interface given by the software is compiled by Fortran language on the Visual Studio platform. Finally, the generated dynamic link file is linked with ABAQUS software to realize the application of the nonlinear interface unit. The operation flow chart is shown in Figure 23.

### 5.4. Interface Model Validation

#### 5.4.1. Static Shear Model Verification

Based on the specific parameters of the polymer–silt clay interface test, the numerical model of the polymer–soil interface was established by ABAQUS, as shown in Figure 24. In the vertical stress conditions of 100 kPa, 200 kPa, 300 kPa and 400 kPa, the simulation values and test results are shown in Figure 25.

It can be seen from Figure 25 that the numerical simulation results using nonlinear interface elements are in good agreement with the experimental results. Under the vertical stress conditions of 100 kPa, 200 kPa, 300 kPa and 400 kPa, the average relative error is 10.74%. The average error of the simulation result meets the reasonable range. Therefore, it is reasonable and feasible to use the above nonlinear interface element to simulate the static properties of the polymer–silty clay interface.

#### 5.4.2. Dynamic Shear Model Validation

In the vertical stress conditions of 100 kPa, 200 kPa and 300 kPa, the polymer–soil interface dynamic model simulation and test results are shown in Figure 26.

It can be seen from Figure 26 that the dynamic shear stress–displacement relationship curves of the polymer–soil cyclic shear simulation results and test results are basically consistent, showing nonlinear properties. The average error between the simulation results is 14.91%, which is in a reasonable range. Therefore, it is reasonable and feasible to use the above nonlinear interface element to simulate the dynamic properties of the polymer–soil interface.

## 6. Conclusions

The polymer anti-seepage wall is a new type of anti-seepage reinforcement technology for earth–rock dam engineering with the advantages of being fast, efficient and durable, with good seismic and anti-cracking properties, etc., and is currently applied in reinforcement projects in many places. However, due to the “root-like” contact state of the new polymer–soil material and the complex stress state of the wall–dam interface, the interface properties between the wall and the dam are unclear, which limits the promotion and application of polymer anti-seepage walls. Therefore, based on the original designed polymer–soil interface test mould, the SRS-150 dynamic ring shear instrument was used to conduct ring shear test research on the static and dynamic properties of the polymer–soil interface, and the following research findings and conclusions were drawn:

(1)The interface properties of polymer–silty clay are better than those of the polymer–silty interface, concrete–silty clay interface and plain soil interface, which are reflected in the higher shear strength, greater adhesive force of the polymer–silty clay interface and more obvious improvement in the shear properties of the earth–rock dam. As the viscosity of the soil increases, the adhesive force and internal friction angle of the interface increase as well, and the silty clay contains more clay particles compared to the silty clay, which enhances the adsorption and adhesion of the interface between the polymer and the nearby soil, resulting in the peak shear stress being further improved. At the same time, compared to conventional concrete-based anti-seepage structures, the polymer bonds with the soil more closely due to adhesive and consolidation effects, which leads to the shear resistance of the polymer–soil being better than the traditional concrete materials.(2)The bonding state, polymer density and vertical stress affect the static properties of the polymer–silty clay interface, and the specific performances are as follows: the shear resistance of the polymer–silty clay interface in the bonded state is better than that in the non-bonded state; meanwhile, the peak shear stress of the polymer–silty clay interface increases with the increases in polymer density and vertical stress. The influence of the bonding state and polymer density on the static properties of the interface is mainly through changing the adhesion between the soil and the structure to increase the adhesive force and internal friction angle of the interface, and the increase in polymer density enhances the expansion anchorage force of polymer materials and strengthens the shear strength of the interface. When the vertical stress increases, the compactness of the soil near the interface of polymer–silty clay also increases, and the frictional resistance between soil particles increases accordingly as well, which drives more soil particles to flip, roll and rearrange to increase the peak shear stress of the interface.(3)The polymer density, shear displacement amplitude and vertical stress affect the dynamic properties of the polymer–silty clay interface, and the specific performances are as follows: the hysteresis loop area and dynamic shear stress of the polymer–silty clay interface increase with the increase in polymer density, shear displacement amplitude and vertical stress. The effect of polymer density on the dynamic properties of the interface is mainly through enhancing the shear modulus and shear strength of the polymer materials; the effect of shear displacement amplitude on the dynamic properties of the interface is mainly through acting on the soil near the interface, resulting in a more orderly arrangement of soil particles near the interface of polymer–silty clay, so that the energy absorbed and dissipated increases accordingly, which further improves the shear stress; the influence of vertical stress on the dynamic properties of the interface is mainly through enhancing the frictional restraint between the soil particles near the interface and squeezing the soil near the interface to increase the peak shear stress of the interface hysteresis loop.(4)Polymer density, shear displacement amplitude and vertical stress have similar effects on the shear stiffness of the polymer–silty clay interface: the shear stiffness of the interface increases with the increases in polymer density, shear displacement amplitude and vertical stress. In addition, the damping ratio of the polymer–silty clay interface is negatively correlated with polymer density and vertical stress, while the damping ratio of the polymer–silty clay interface is sensitive to the shear displacement amplitude, and the damping ratio corresponding to the same number of cycles is positively correlated with the shear displacement amplitude.(5)According to the interface properties of polymer–silty clay, the construction process of the hyperbolic constitutive model is systematically expounded by using mathematical formulae. Verified by numerical simulation, the test values of the shear strength and shear displacement curves of the static shear model and cyclic shear model have a good fitting degree with the simulation values.

## Figures and Tables

**Figure 1 polymers-15-00634-f001:**
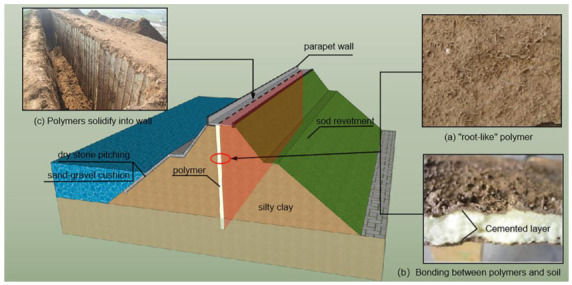
Construction technology of polymer anti-seepage wall.

**Figure 2 polymers-15-00634-f002:**
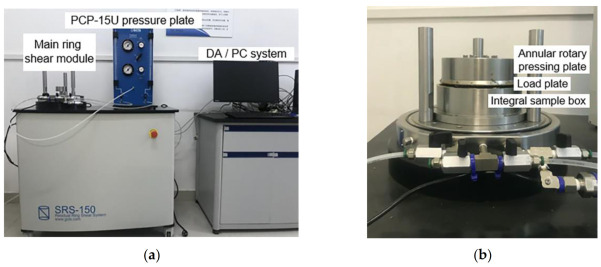
SRS-150 dynamic ring shear test device: (**a**) the PCP-15U pressure plate and the DA/PC system; (**b**) the main ring shear module.

**Figure 3 polymers-15-00634-f003:**
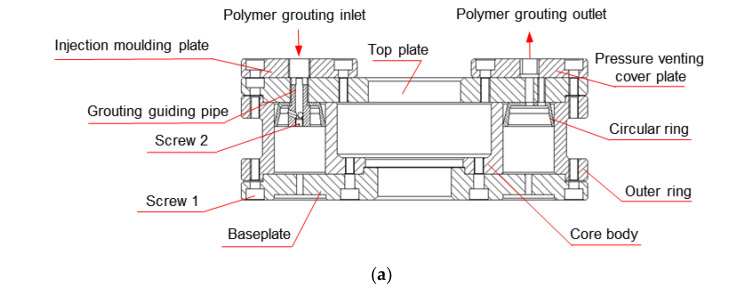
Polymer–soil sample mould: (**a**) section plan of the mould; (**b**) schematic diagram of the mould; (**c**) perspective drawing of the mould.

**Figure 4 polymers-15-00634-f004:**
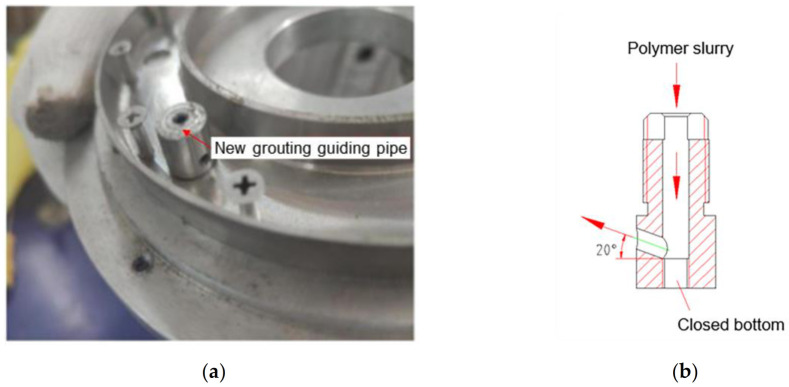
Original design grouting guiding pipe: (**a**) grouting guiding pipe; (**b**) grouting guiding pipe profile.

**Figure 5 polymers-15-00634-f005:**
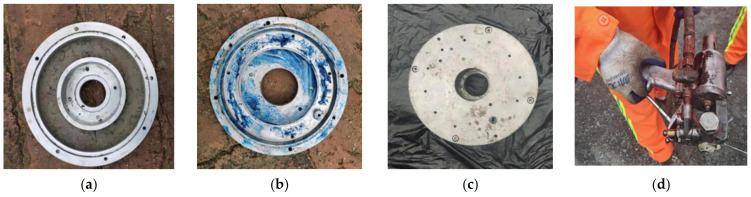
Polymer grouting process: (**a**) filling the soil sample in the core; (**b**) applying lubricating fluid to the ring; (**c**) fastening the connection between the core and the ring; (**d**) grouting into the mould.

**Figure 6 polymers-15-00634-f006:**
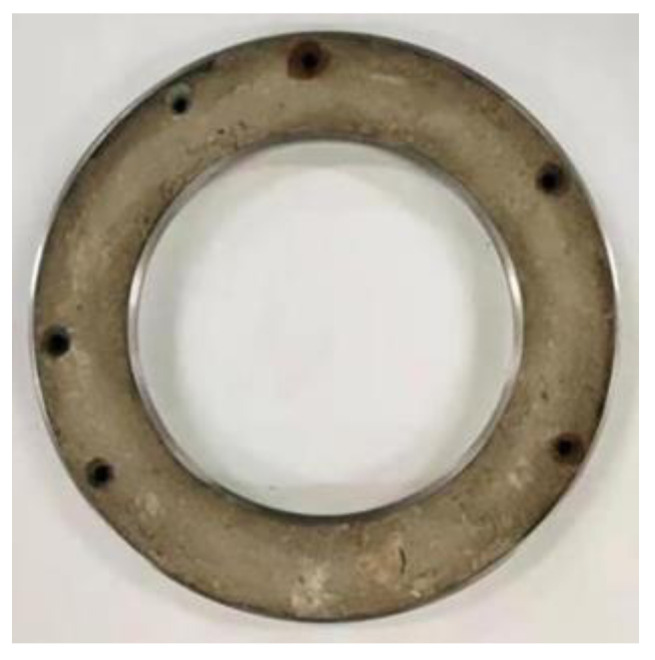
Interface sample after grouting.

**Figure 7 polymers-15-00634-f007:**
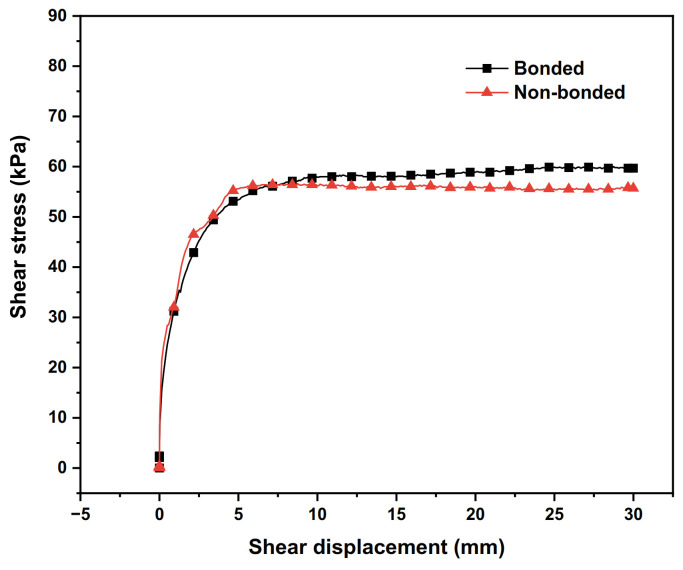
Shear stress–shear displacement curves of different interface bonding states.

**Figure 8 polymers-15-00634-f008:**
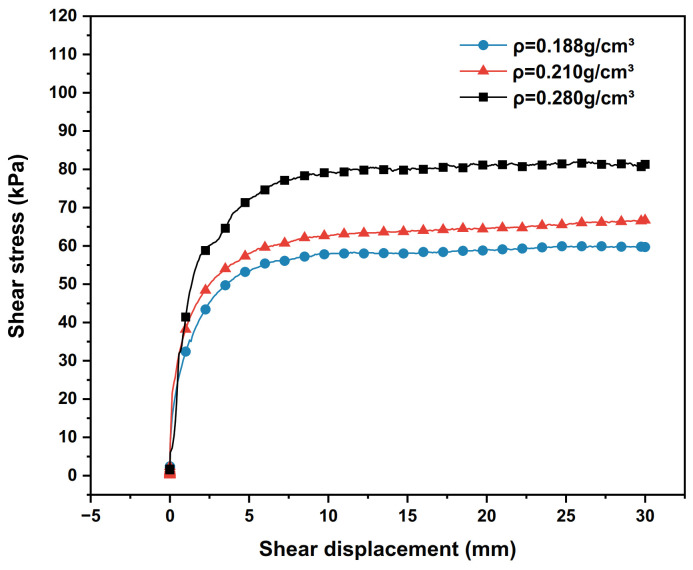
The relationships between the shear stress and the shear displacement of the interface under different polymer densities.

**Figure 9 polymers-15-00634-f009:**
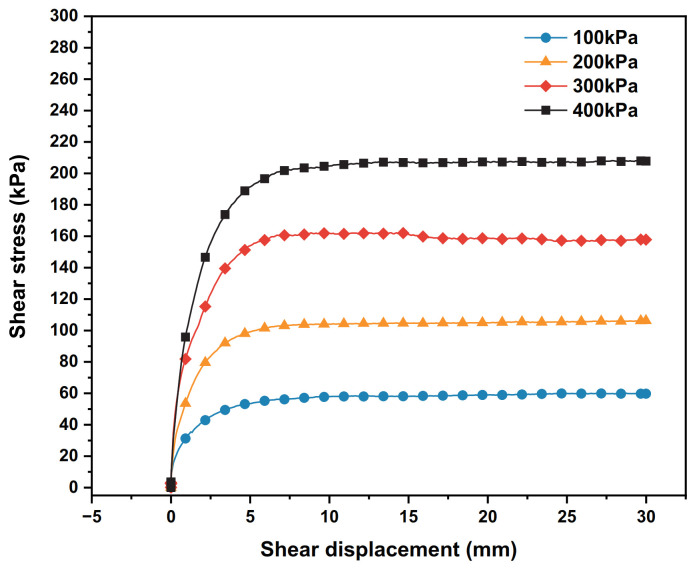
The relationships between the shear stress and shear displacement of the interfaces under different vertical stresses.

**Figure 10 polymers-15-00634-f010:**
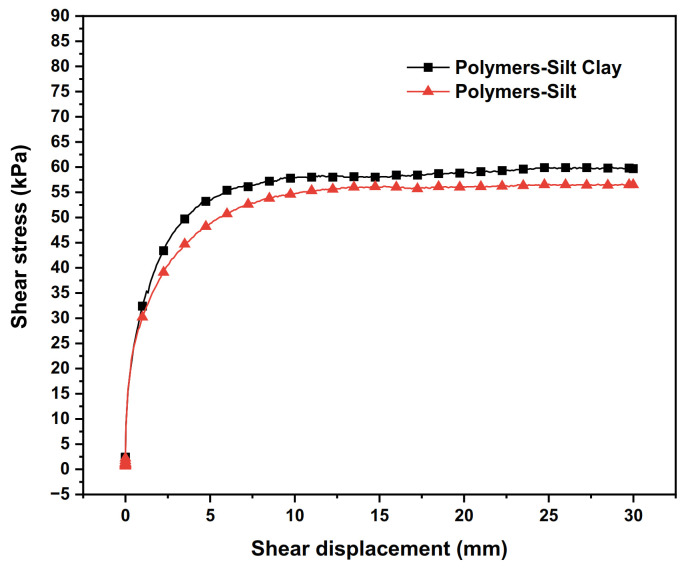
The shear stress–shear displacement relationship curves of the polymer–different dam materials interfaces.

**Figure 11 polymers-15-00634-f011:**
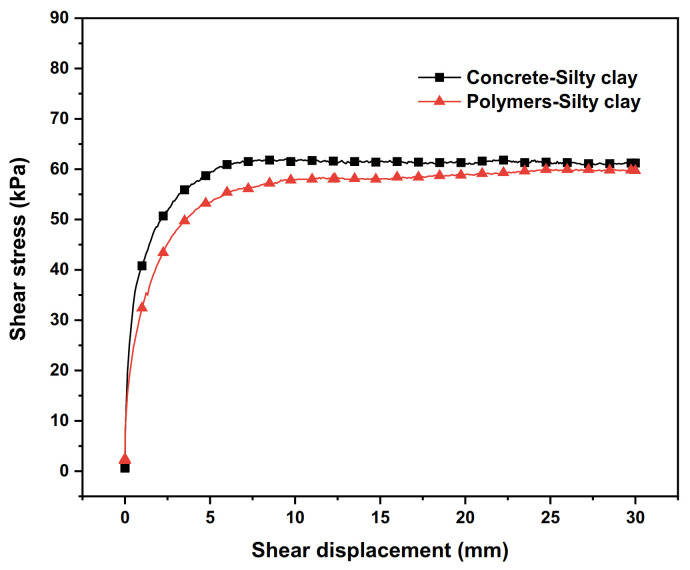
Shear stress–shear displacement relationship curves of different impermeable body–soil interfaces.

**Figure 12 polymers-15-00634-f012:**
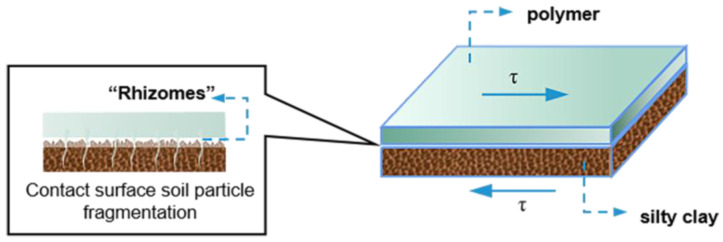
Breakage of soil particles on the interface of polymer–silty clay.

**Figure 13 polymers-15-00634-f013:**
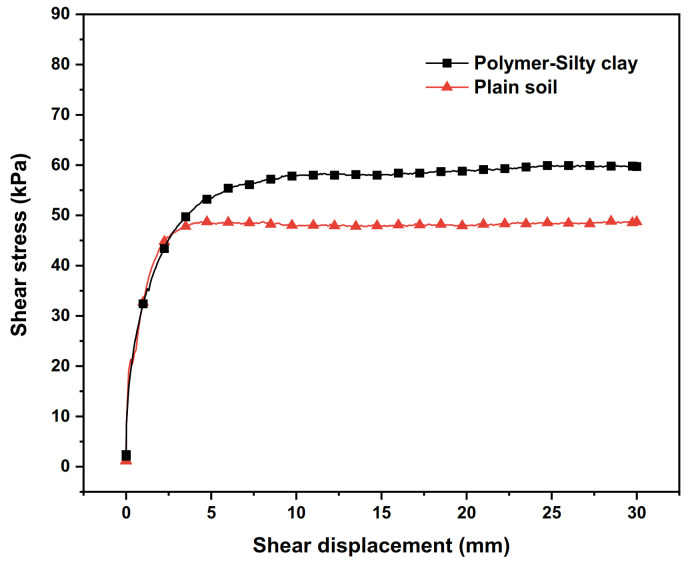
The relationships between shear stress and shear displacement of plain soil and polymer–silty clay.

**Figure 14 polymers-15-00634-f014:**
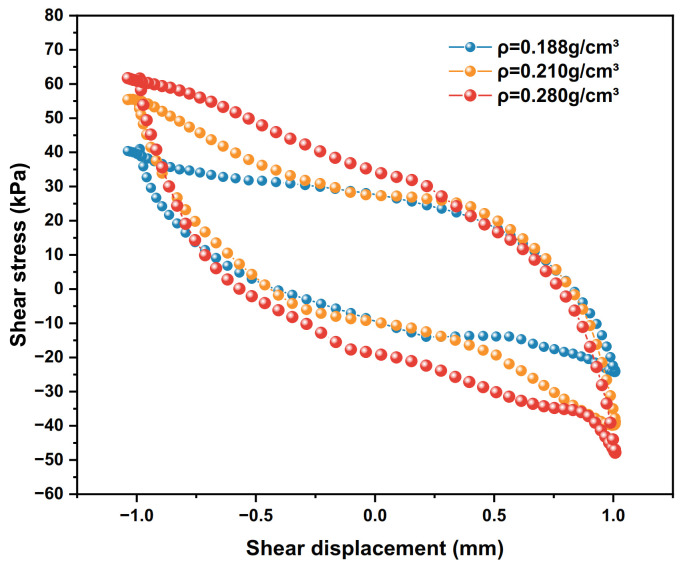
The shear stress–shear displacement relationship curves of different polymer densities.

**Figure 15 polymers-15-00634-f015:**
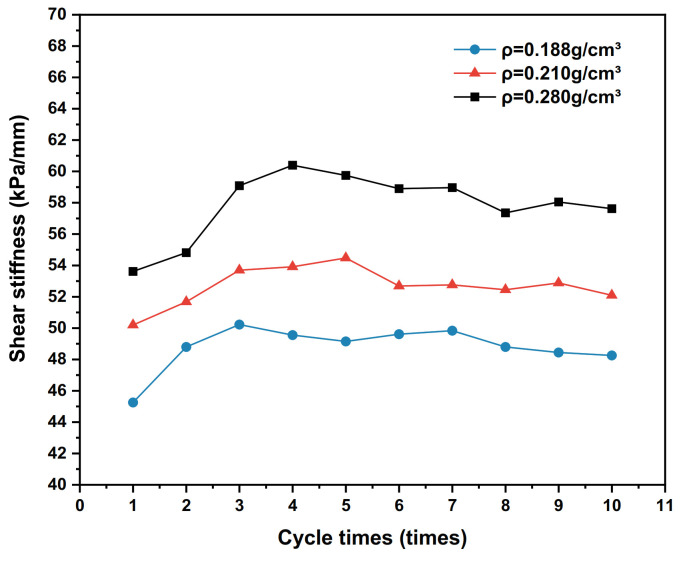
Development trends of shear stiffness with different densities.

**Figure 16 polymers-15-00634-f016:**
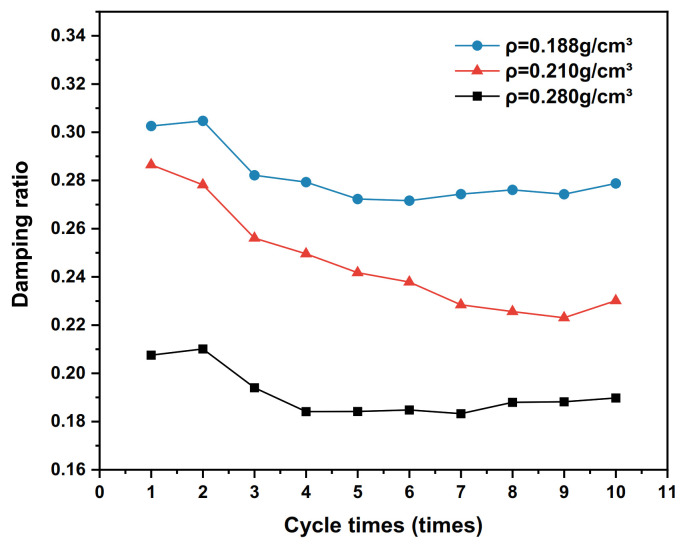
Development trends of damping ratios with different densities.

**Figure 17 polymers-15-00634-f017:**
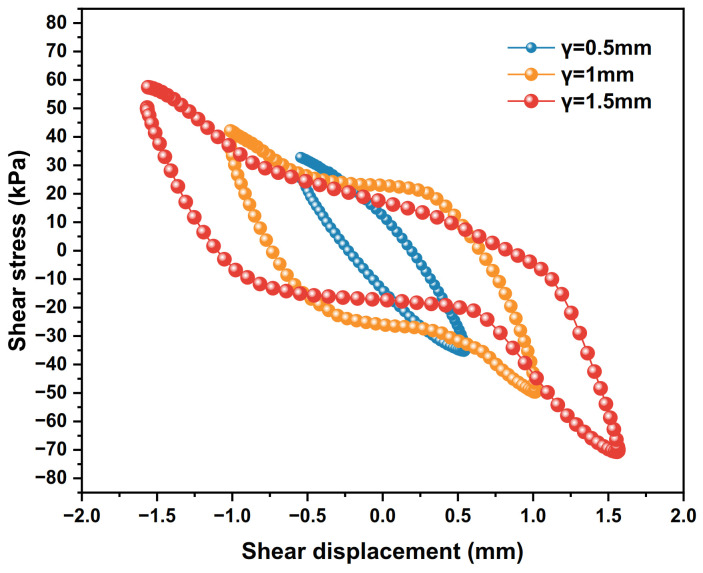
The curves of shear stress and shear displacement under different shear displacement amplitudes.

**Figure 18 polymers-15-00634-f018:**
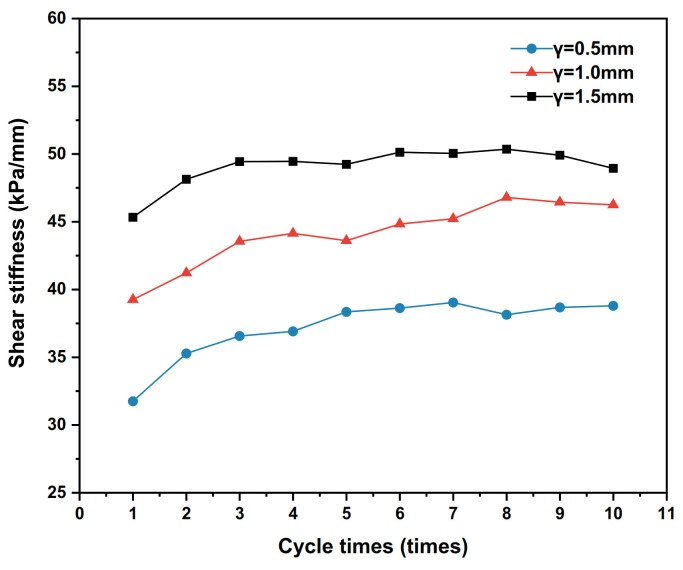
Development trends of shear stiffness with different shear amplitudes.

**Figure 19 polymers-15-00634-f019:**
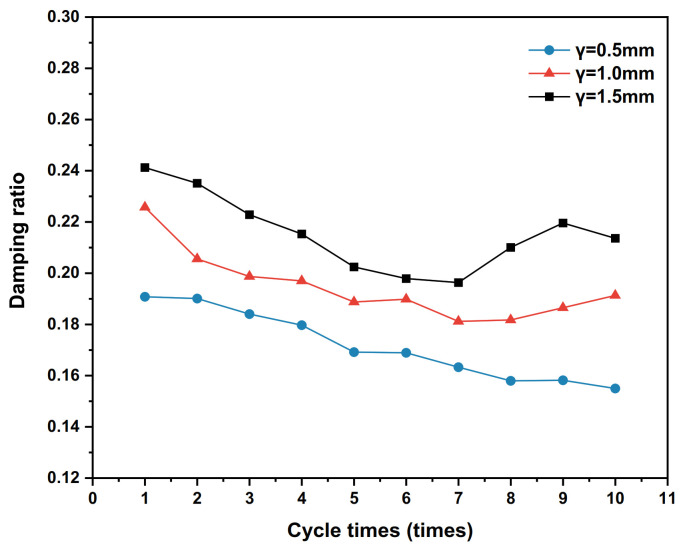
Development trends of damping ratios with different shear amplitudes.

**Figure 20 polymers-15-00634-f020:**
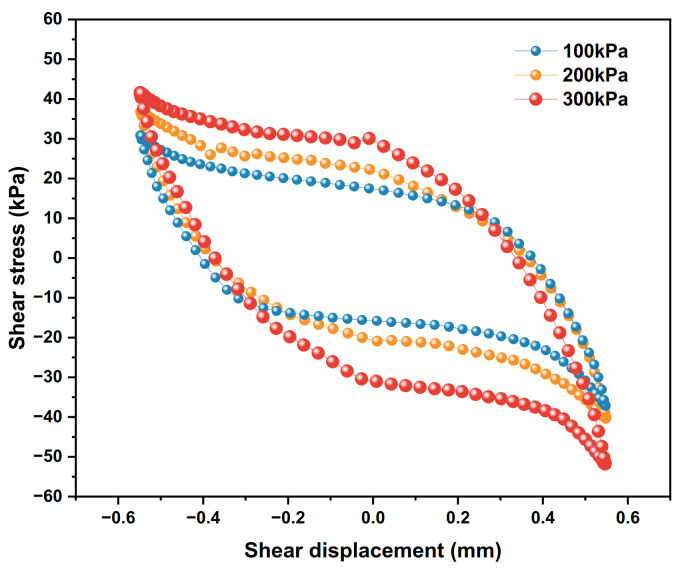
Shear stress–shear displacement relationship curves under different vertical stresses.

**Figure 21 polymers-15-00634-f021:**
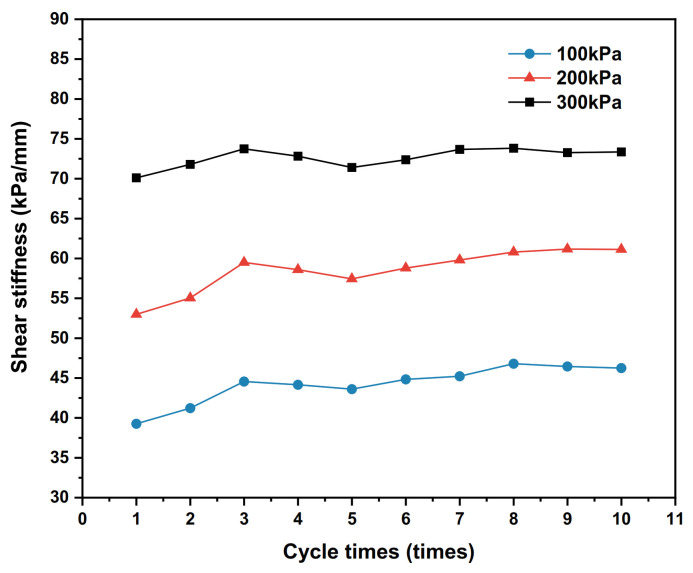
Development trends of shear stiffness under different vertical stresses.

**Figure 22 polymers-15-00634-f022:**
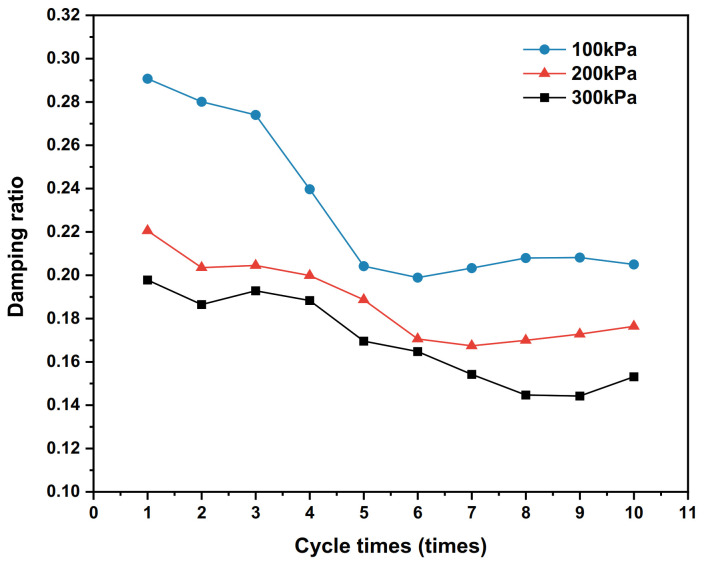
Development trends of damping ratios with different vertical stresses.

**Figure 23 polymers-15-00634-f023:**
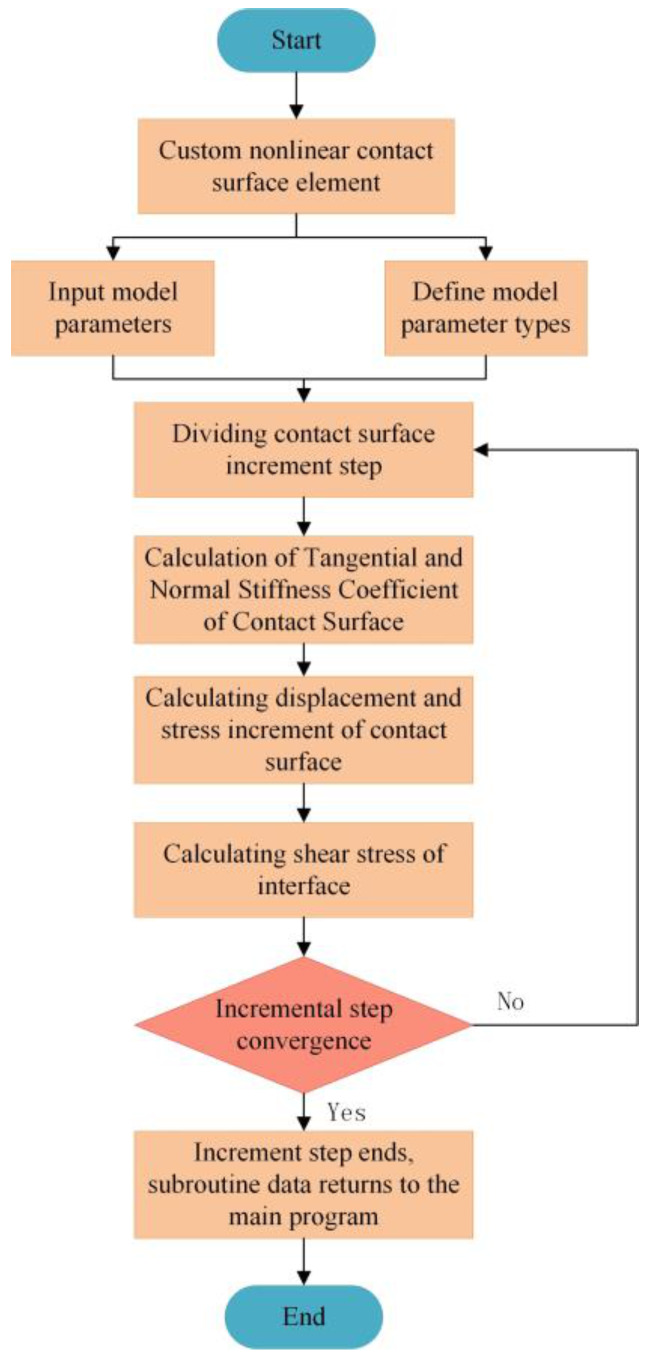
Operation flow chart.

**Figure 24 polymers-15-00634-f024:**
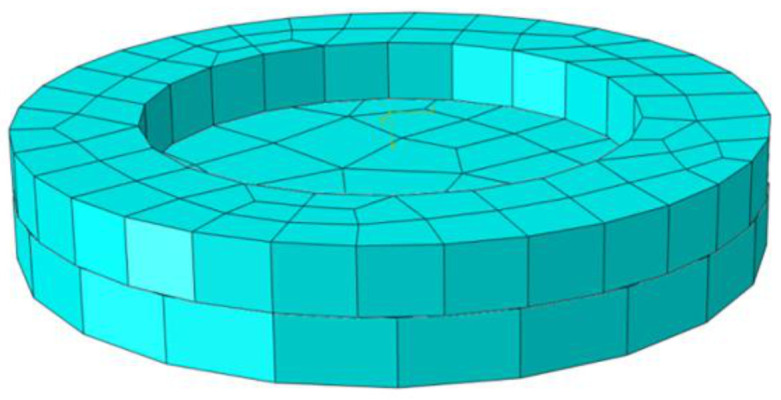
Numerical model of interface.

**Figure 25 polymers-15-00634-f025:**
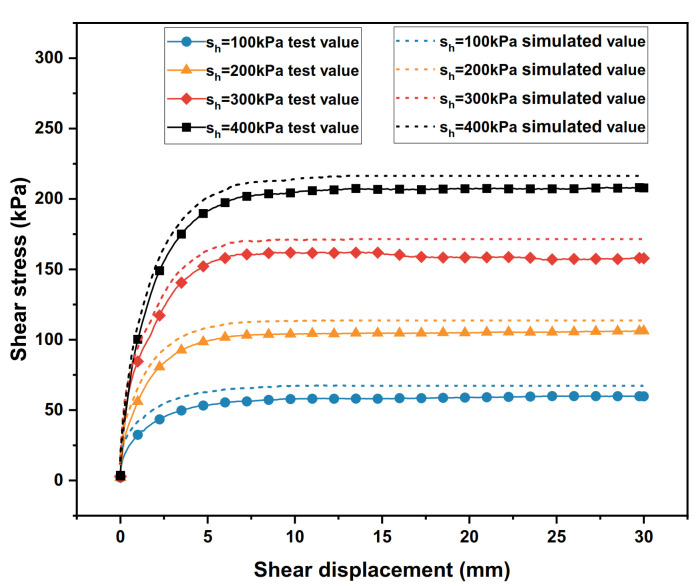
Shear stress–displacement curves of interface static shear simulated values and test values.

**Figure 26 polymers-15-00634-f026:**
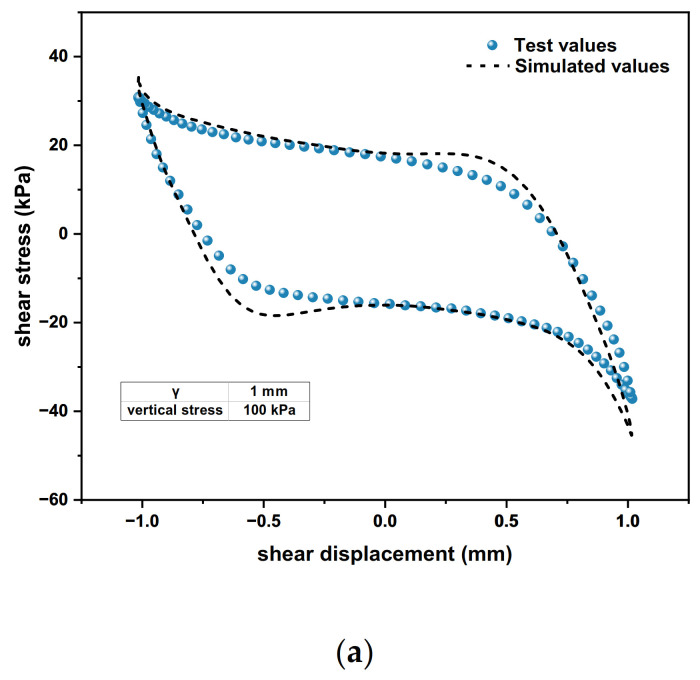
Dynamic shear stress–displacement relationship curves of polymer–soil cyclic shear simulation values and test values: (**a**) vertical stress = 100 kPa; (**b**) vertical stress = 200 kPa; (**c**) vertical stress = 300 kPa.

**Table 1 polymers-15-00634-t001:** Main parameters of SRS-150 test device.

Main Parameters	Value	Unit
exercise frequency	0–5	Hz
shear rate	0.0008–314	mm/min
consolidation stress	0–1000	kPa
vertical displacement	0–50	mm
sample inner diameter	100	mm
sample outer diameter	150	mm
cross-sectional area	98.16	cm^2^

**Table 2 polymers-15-00634-t002:** Polymer technical indicators.

Item	Technical Indicators
Appearance	Light brown transparent liquid
Viscosity/(mPa·s)	60–600
Induced coagulation time/s	10–13000
Expansion ratio/%	350–1000
Permeability coefficient k(cm/s)	1 × 10^−9^
Elastic modulus E (MPa)	58.11
Poisson’s ratio μ	0.26

**Table 3 polymers-15-00634-t003:** Basic physical properties of soil used in experiment.

Specific Gravity Gs	Void Ratio e	Water Content Ω (%)	Permeability Coefficient k (cm/s)	Elastic Modulus E (MPa)	Poisson’s Ratio μ	Cohesion C (kPa)	Internal Friction Angle φ (°)
2.7	0.804	21	2.3 × 10^−5^	37.2	0.35	22.2	11.3

**Table 4 polymers-15-00634-t004:** Static shear test schemes of polymer–soil interface.

Test Type	Test Number	Type of Interface	Water Content	Shear Rate	Density	Vertical Stress
			(%)	(mm/min)	(g/cm^3^)	(kPa)
static shear	A-1	polymer–silty clay (bonding)	21	1	0.188	100
	A-2	polymer–silty clay (non-bonded)				
	B-1	polymer–silty clay	21	1	0.188	100
	B-2				0.210	
	B-3				0.280	
	C-1	polymer–silty clay	21	1	0.188	100
	C-2					200
	C-3					300
	C-4					400
	D-1	polymer–silty clay	21	1	0.188	100
	D-2	polymer–silt				
	E-1	polymer–silty clay	21	1	0.188	100
	E-2	concrete–silty clay			——	
	F-1	polymer–silty clay	21	1	0.188	100
	F-2	plain soil			——	

**Table 5 polymers-15-00634-t005:** Dynamic shear test schemes of polymer–soil interface.

Test Type	Test Number	Type of Interface	Density	Cycle Amplitude	Vertical Stress
			(g/cm^3^)	(mm)	(kPa)
cyclic shear	G-1	polymer–silty clay	0.188	1	100
	G-2		0.210		
	G-3		0.280		
	H-1	polymer–silty clay	0.188	0.5	100
	H-2			1	
	H-3			1.5	
	I-1	polymer–silty clay	0.188	1	100
	I-2				200
	I-3				300

**Table 6 polymers-15-00634-t006:** Shear strength indexes of interfaces in different bonding states.

Type of Interface	c/kPa	φ/°
polymer–silty clay (bonding)	9.05	25.55
polymer–silty clay (non-bonded)	4.85	25.13

**Table 7 polymers-15-00634-t007:** The shear strength indexes of the interfaces with different polymer densities.

Type of Interface	c/kPa	φ/°
polymer–silty clay (ρ = 0.188 g/cm^3^)	9.05	25.55
polymer–silty clay (ρ = 0.210 g/cm^3^)	15.20	26.61
polymer–silty clay (ρ = 0.280 g/cm^3^)	34.90	27.11

**Table 8 polymers-15-00634-t008:** The shear strength indexes of the interfaces of different dam materials.

Type of Interface	c/kPa	φ/°
polymer–silty clay	9.05	25.55
polymer–silt	7.75	21.45

**Table 9 polymers-15-00634-t009:** Shear strength indexes of the interfaces of different impermeable bodies.

Type of Interface	c/kPa	φ/°
polymer–silty clay	9.05	25.55
concrete–silty clay	5.50	28.90

**Table 10 polymers-15-00634-t010:** The shear strength indexes of polymer–silty clay interface and plain soil.

Type of Interface	c/kPa	φ/°
polymer–silty clay	9.05	25.55
plain soil	8.63	24.84

**Table 11 polymers-15-00634-t011:** Static model parameters of polymer–soil interface.

Interface	$K_{x}$	$K_{y}$	$R_{f}$	n	δ	$\gamma_{w}$	$P_{a}$
polymer–soil	1137	1137	0.89	0.45	26.57	10	100

**Table 12 polymers-15-00634-t012:** Dynamic model parameters of polymer–soil interface.

Interface	$K_{x}$	$K_{y}$	$R_{f}$	*n*	δ	$\gamma_{w}$	$P_{a}$	$\lambda_{0}$	$\lambda_{ult}$	*k*	$\alpha_{1}$	$\alpha_{2}$
Polymer–soil	1137	1137	0.89	0.45	26.57	10	100	0.31	0.27	1.25	1.84	0.01

## Data Availability

The data presented in this study are available within this article.

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
