# Peer review of "Static and Dynamic Properties Study on Interface between New Polymer Materials and Silty Clay Based on Ring Shear Tests"

_polymers, 2023, doi:10.3390/polym15030634_

Round 1

Reviewer 1 Report

The authors study characteristics of the static and cyclic shear at the substrate-polymer interface and corroborate their experimental results with simulations. The study is valuable in practical terms for engineering anti-seepage reinforcements in dams and dykes.

I would definitely recommend the manuscript for publication after the authors address the following comments and questions.

1. The manuscript should be carefully proofread so that the language flaws be corrected.

2. From L. 206-275 and 580-581, it remains unclear which exactly substrates were compared in terms of shear properties. The authors mix the terms "silty clay", "silt" and "silty soil", whereas in the Abstract, only polymer-soil interface is mentioned. Please revise.

3. How is the state of bonding (i.e. bonding vs. non-bonding) of polymer to the substrate controlled?

4. The authors claim that "the development of the interface damping ratio under the three vertical stresses tends to a fixed value." However, this is not seen in Fig. 22 as the damping factor curves demonstrate quite irregular trends.

Reviewer 2 Report

In this paper, the static and dynamic properties of the polymer-soil interface are explored, as are the influence laws and related mechanisms of different factors on the polymer-soil interface. At the same time, a hyperbolic constitutive model of the polymer-soil interface was established, and the validity of the model was verified by comparing the numerical simulation with the experimental results.

After finishing tests, the conclusion was that, compared to conventional concrete-based anti-seepage structures, the polymer will bond with the soil more closely due to adhesive and consolidation effects, which leads to the shear resistance of the polymer soil being better than the traditional concrete materials by resulting in a more orderly arrangement of soil particles near the interface of polymer-silty clay. The shear stiffness of the interface also increases with the increase of polymer density.

Given that the research objectives were clearly stated and the results were well-presented, in accordance with the standards of a scientific article, the reviewer considers that the paper could be published in the journal Polymers.

Reviewer 3 Report

The manuscript entitled "Static and dynamic properties study on interface between new polymer materials and silty clay based on ring shear tests" is mainly focused on the static and dynamic ring shear tests of the polymer-soil interface.

 The abstract section is detailed with a concise description of the possible application. Is this material required in this section? Identical information is provided in the introductory chapter with detailed explanations and illustrations. I would suggest that the authors in this chapter should focus more on the materials used and the research done, rather than the potential application, which seems not to be allowed in more developed countries. Besides, with the increasing amount of plastic in the global environment, I do not think that such kind of technology is environmentally acceptable.

 Are there other areas of use for isocyanate and polyol components on a smaller scale? This should probably be mentioned in the introduction section as well.

 Isocyanate is a toxic material. Is it logical and safe to use this material in significant quantities for big polymer anti-seepage walls?

 The characteristics of the components used in the synthesis of the polymer should be described in detail: supplier, purity, degree of polymerization, etc.

 Unclear numbering of tests in Tables 4 and 5. What do the codes A-1, B-1, C-1, etc. mean? How was the density of each sample determined? Could the authors explain how the polymer density in a specific object would be controlled?

 The authors could pay more attention to the polymer synthesis procedure itself. Is this process so old and clear that it is no longer necessary to pay attention to it? What is the novelty of this work from a synthesis perspective?

 Do the polymer synthesis conditions have an effect on the relationships between the shear stress and shear displacement?

 The adhesive and consolidation effects between the polymer and the soil should be explained from the chemical point of view. What is the chemical composition of soil? In this case, the authors describe possible interaction very vaguely.

 It can be argued that the chemical side of this work is not the most relevant. The conclusions of the obtained results are based on purely mechanical properties of the mixtures. The problem of polymer degradation, which is particularly important in the environment, was not discussed.
